# Growth of non-layered 2D transition metal nitrides enabled by transient chloride templates

Liqiong He[1,2], Jingwei Wang[1,2,3], Zhengyang Cai[4], Ruiting Liu[1,2], Shengnan Li[1,2], Yunhao Zhang[1,2], Zhi-Yuan Zhang[1,2], Jiarong Liu[1,2] & Bilu Liu [1,2] ✉

2D transition metal nitrides (TMNs) have attracted significant attention due to their magnetic, electrical, and chemical properties at atomic thickness. However, the synthesis of 2D TMNs is still challenging, due to their strong isotropic metal-nitrogen bonding networks. Here, we report a universal synthesis of non-layered 2D TMN family by using corresponding metastable metal chlorides as transient templates. This approach takes advantage of the layered structures and low conversion energy barriers of transition metal chlorides (TMCls) to grow 2D TMNs. Fifteen types of 2D TMNs and their alloys were synthesized, demonstrating the versatility of this method. The 2D TMN family exhibits tunable magnetic characteristics ranging from antiferromagnet to hard magnet, which can be modulated by their composition. This work overcomes previous synthesis limitations, thus offering a pathway to explore fundamental properties of 2D TMNs and accelerate their applications.

Transition metal nitrides (TMNs) represent a class of materials with impressive properties, including ultrahigh mechanical strength, remarkable thermal stability, tunable magnetism, and superconductivity[1-4]. Recent advances in two-dimensional (2D) materials have spurred interest in reshaping TMNs into ultrathin 2D forms to unlock enhanced functionalities[5-8]. For instance, owing to unsaturated dangling bonds and low-coordination atoms, the exposed surfaces endowing 2D TMNs with excellent catalytic performance and chemisorption capabilities in energy-related applications[4,9-13]. Notably, under electron and phonon confinement at the 2D limit, these ultrathin TMNs exhibit properties distinct from their bulk counterparts, offering opportunities for fundamental studies in magnetism[5,14-16], electronic transport[5,8], optical properties[8], etc. Additionally, due to their atomic-level thickness, 2D TMNs could exhibit high flexibility, making them promising building blocks for future 2D electronic, spintronics, and optoelectronic devices[17-21]. To achieve these property explorations

and applications, the prerequisite is the universal and controllable synthesis of 2D TMNs.

Different from other nonlayered materials such as transition metal oxides (TMOs, e.g., $Fe_2O_3$) and transition metal chalcogenides (e.g., FeS), TMNs shows special isotropic hybrid metallic-covalent interactions between metal and nitrogen atoms in three dimensions[22-26]. This isotropy impedes the formation of thin 2D TMNs by using the established method in synthesizing other non-layered 2D materials[2,3,27-29]. Currently, approaches such as selective etching of MAX phases[17,30,31], salt-templated growth[32,33], and 2D template conversion[34,35], have been developed to produce certain free-standing 2D TMN materials. For example, Urbankowski et al have fabricated 2D $Ti_4N_3$-based MXene by using the molten fluoride salts to etch Al atoms from $Ti_4AlN_3$ powder precursor[30]. Note that the produced materials contain unavoidable surface terminal groups, which influence the understanding of the intrinsic properties of pure 2D $Ti_4N_3$. In another work, Jin et al have obtained layered $W_2N_3$ nanosheets by

[1]Shenzhen Geim Graphene Center, Shenzhen Key Laboratory of Advanced Layered Materials for Value-added Applications, Institute of Materials Research, Tsinghua Shenzhen International Graduate School, Tsinghua University, Shenzhen, PR China. [2]Key Laboratory of Electrocatalytic Materials and Green Hydrogen Technology of Guangdong Higher Education Institutes, Tsinghua Shenzhen International Graduate School, Tsinghua University, Shenzhen, PR China. [3]School of Flexible Electronics, Sun Yat-sen University, Shenzhen, PR China. [4]School of Integrated Circuits, Jiangnan University, Wuxi, Jiangsu, PR China. ✉e-mail: bilu.liu@sz.tsinghua.edu.cn

ammonization of 2D Na$_2$W$_4$O$_{13}$ salt-templates with similar lattice symmetry, while this method remains limited to a few 2D TMNs since the lattice matching requirement between salt and target materials[10]. Although the 2D template conversion approach shows promise for the universal growth of high-quality pristine 2D TMNs, selecting suitable template materials to avoid the long nitriding time caused by etching is critical[34,35]. Transition metal chlorides (TMCls) therefore is a potential choice since their low conversion energy, while their high-temperature instability makes this process practically unachievable[36,37]. How to stabilize and nitrogenize these metastable templates is critical for the successful growth of 2D TMNs.

Here, we develop a transient chloride template assisted reverse-thermal-field (RTF) strategy for the universal synthesis of a library of 2D TMNs and their alloys. We use TMCls as templates, taking advantage of their layered structure and low conversion energy barriers to corresponding nitrides. By applying a spatially RTF method during

nitridation in a short time, we directly transform metastable TMCls templates into ultrathin TMNs with preserved integrity and single-crystallinity. Through changing the type of precursor and growth temperature, we have grown 7 types of 2D TMNs and 8 types of 2D TMN alloys. The high-resolution transmission electron microscopy (HRTEM) images confirm the structure and composition of materials. Magnetic measurements show that the 2D TMNs exhibit magnetic behaviors different from their bulk counterparts, and alloying is proven to be an effective way to modulate their magnetism.

## Results

### Growth of 2D TMNs by the reverse-thermal-field CVD method

As shown in Fig. 1a, compared to common transition metal compounds like oxides and sulfides, TMCls possess two key advantages which make them promising templates to grow 2D TMN. First, TMCls show the lowest solid-to-solid conversion energy to TMNs at high

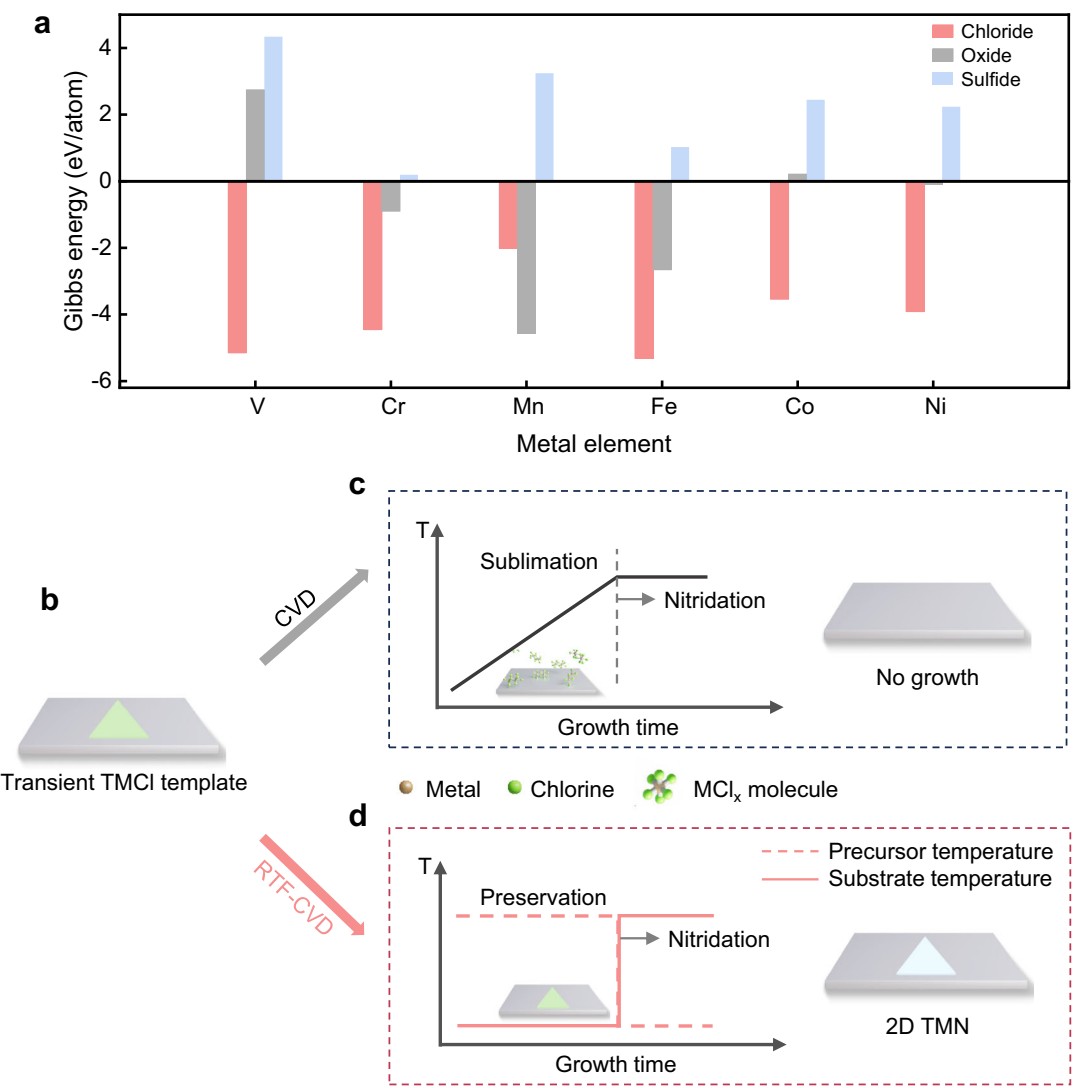

**Fig. 1 | Design and growth process of the reverse-thermal-field (RTF) chemical vapor deposition (CVD) method. a** Calculations of the Gibbs energy barriers of the solid-to-solid conversion from transition metal chlorides, oxides, and sulfides to the transition metal nitrides (TMNs) counterparts at 1000 K. The positive energy means the conversion is theoretically unfavorable, while the negative energy means it is theoretically favorable. **b** Schematic of the metastable 2D transition metal chloride (TMCl) template. **c** Scheme to illustrate the temperature program and growth result of the traditional CVD method. The nitridation process starts when

the temperature reaches the thermodynamically supported conversion temperature. And the TMCl is sublimated as MCl$_x$ molecules before the nitridation process. **d** Scheme to illustrate the temperature program and growth result of the RTF CVD method. The dashed and solid lines represent the temperature of the precursor and substrate, respectively. The nitridation process starts when the thermal field is reversed. The detailed process is shown in the Methods and Supplementary Information.

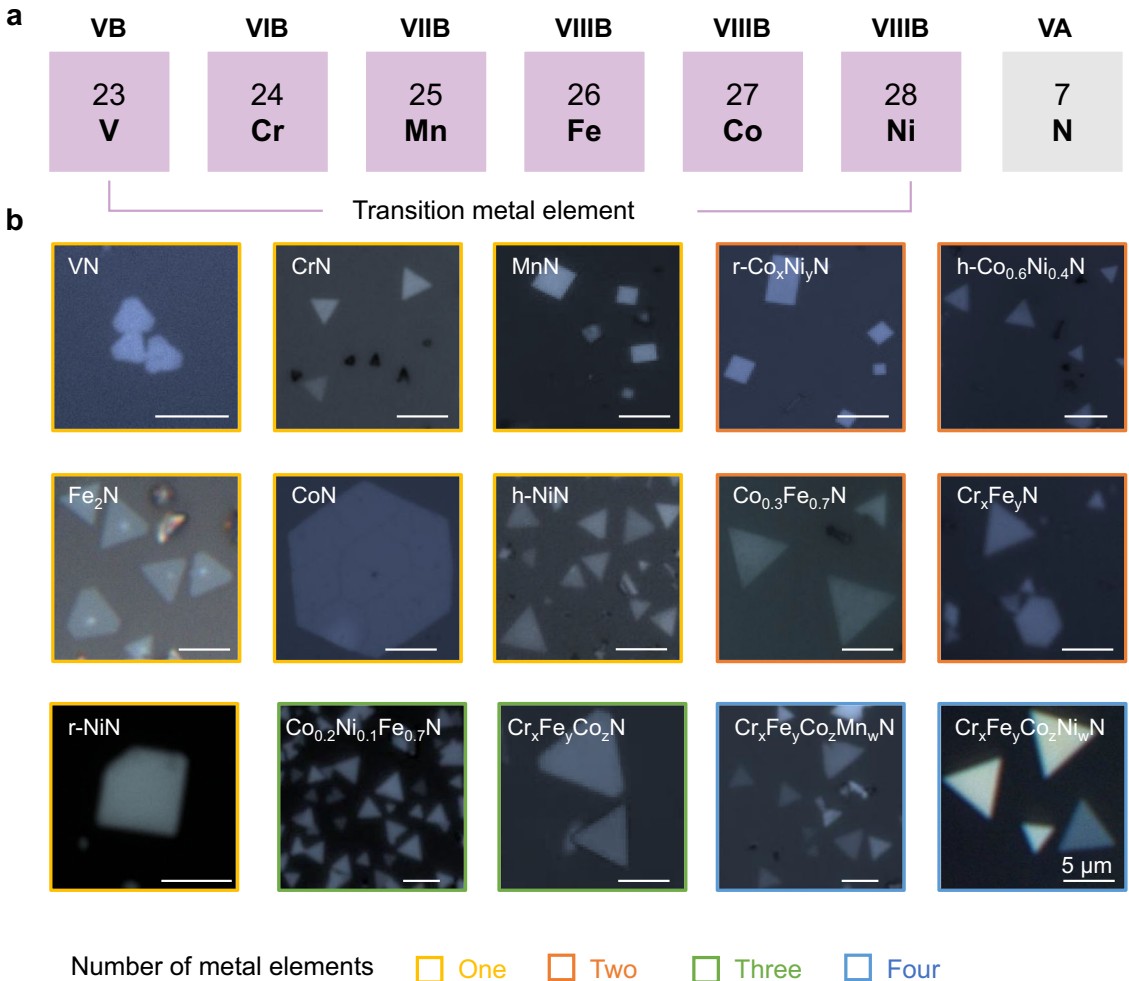

**Fig. 2 | Universal growth of 2D TMNs and their alloys by the RTF method. a** The periodic table of elements shows the transition metal elements involved in this work. **b** Optical microscopy images of the 15 types of 2D TMNs and alloys. All the scale bars are 5 μm. The materials based on one, two, three, and four types of metal elements are highlighted in yellow, orange, green, and blue, respectively.

temperatures, which reduces the conversion time and thus inhibits crystal etching caused by $NH_3$ and $H_2$ during the conversion process. Second, unlike the transition metal oxides and sulfides which mainly show nonlayered structures, most of the TMCls exhibit a layered vdW structure (Supplementary Fig. 1). However, due to their high volatility and hygroscopicity, the 2D TMCl templates cannot exist stably at high temperatures, making the direct use of their chloride template unapplicable. Indeed, Supplementary Fig. 2 shows the optical microscopy (OM) images of a 2D TMCl flake exposed in air for 10 s, 20 s, 30 s, 40 s, and 50 s, and Supplementary Fig. 3 shows the OM images of deposited TMCl heated at 650 °C in an Ar atmosphere. The quick fading of the flake demonstrates its poor stability. In addition, as shown in the thermal gravimetric analysis (TGA) curves (Supplementary Fig. 4a), all the sublimation temperatures of TMCls (i.e., $VCl_3$, $CrCl_3$, $MnCl_2$, $FeCl_3$, $CoCl_2$, and $NiCl_2$) are below 200 °C, revealing that the 2D TMCl templates cannot maintain as solid crystal at high temperature. However, the thermodynamically supported conversion temperature from chlorides to nitrides is above their sublimation temperature, which indicates that the solid-to-solid conversion from 2D TMCls to 2D TMNs is difficult (Supplementary Fig. 4b), making the traditional CVD method unapplicable (Fig. 1b, c). Interestingly, we found that the sublimation rate of TMCl in $NH_3$ was greater than its growth rate to nitride when the growth temperature was <700 °C, while it was smaller when the temperature was >700 °C (Supplementary Fig. 5). Therefore,

the key is how to heat the metastable TMCl templates to the required high temperature in a short time.

Based on this, we develop an RTF method in which the fast switching of the thermal field makes it possible to satisfy the above requirements. Fig. 1d and Supplementary Figs. 6–9 illustrate the setup of the furnace and the growth process of RTF method and traditional CVD method (see details in Methods and Supplementary Information). The step 1 stands for the growth of 2D TMCl templates, and the step 2 represents their conversion to the corresponding 2D TMNs. In the step 1, the TMCl precursors were placed at the upstream of the growth tube (Field I), where the growth temperature was 600-800 °C. At the same time, the mica substrates were placed at the downstream of the tube (Field II), where the temperature was about 100-200 °C to grow a 2D TMCl template under Ar. Subsequently, in the step 2, the temperature of the Field I and Field II was reversed in a short time (< 2 s) by moving the furnace while keeping the position of the quartz tube fixed, together with the introduction of $NH_3$. In this step, the 2D TMCl templates were in situ converted to corresponding 2D TMNs when the temperature of Field II quickly reached up to 600-800 °C. At the same time, the temperature of Field I was quickly cooled down to 100-200 °C, inhibiting the extra evaporation of precursors induced contamination. After growth, the temperature of the furnace was cooled down to room temperature (R.T.). Notably, since the TMCls show similar physical and chemical properties, various 2D TMNs were

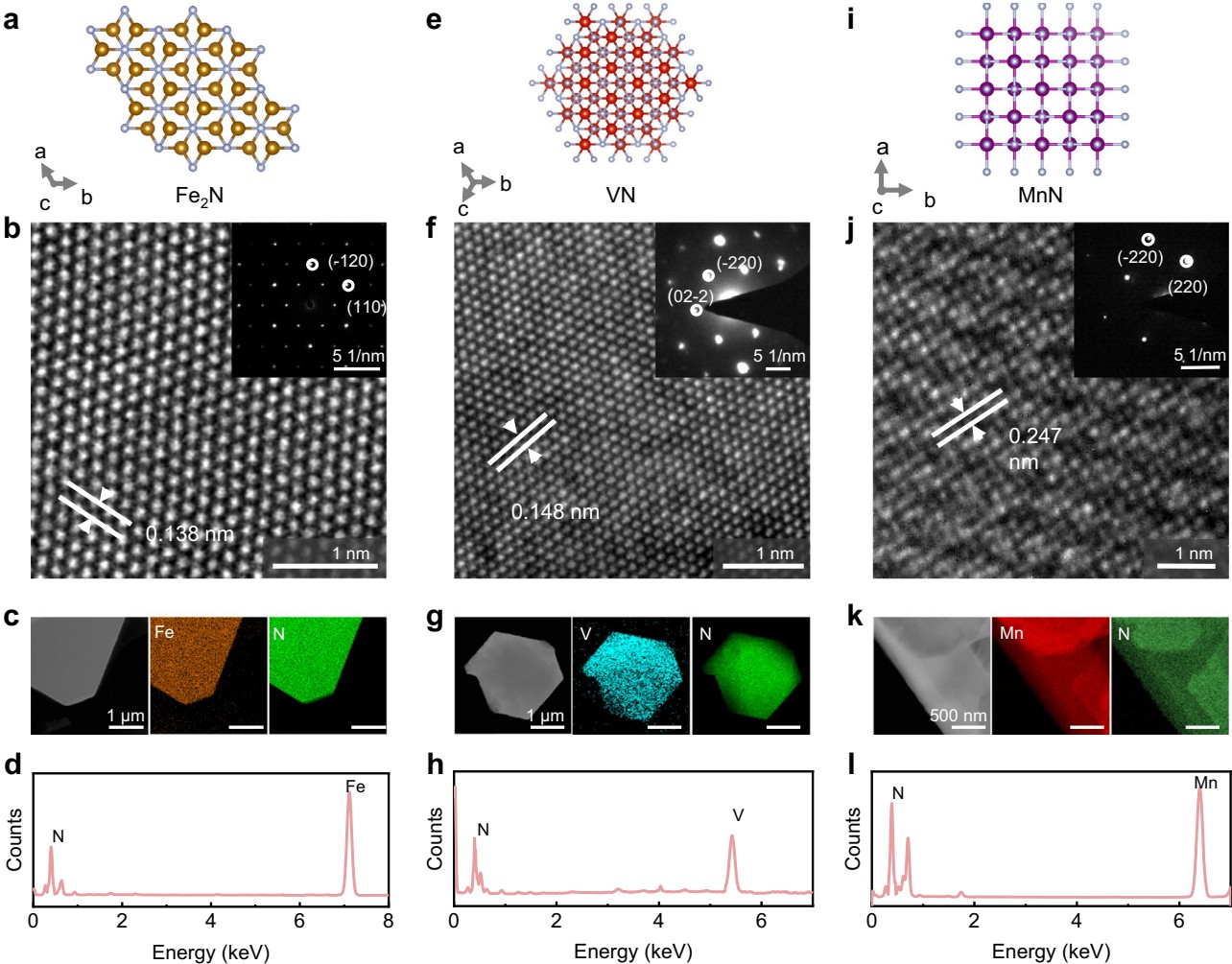

**Fig. 3 | Structural and composition characterization of 2D TMNs. a** Top-view atomic structure of Fe$_2$N. **b** High-resolution transmission electron microscopy (HRTEM) image and selected area electron diffraction (SAED) pattern (inset) of the synthesized 2D Fe$_2$N flake. **c** Low magnification TEM image, energy-dispersive spectroscopy (EDS) element mapping of Fe and N atoms, and (**d**) EDS of the 2D Fe$_2$N flake. **e** Atomic structure of VN from the top view. **f** HRTEM image of the synthesized 2D VN flake. The inset is the corresponding SAED pattern. **g** Low magnification TEM image, elemental EDS mapping of V and N atoms, and **h** EDS of the 2D VN flake. **i** Atomic structure of MnN from the top view. **j** HRTEM image of the synthesized 2D MnN flake. The inset is the corresponding SAED pattern. **k** Low magnification TEM image, elemental EDS mapping of Mn and N atoms, and (**l**) EDS of the 2D MnN flake.

fabricated by changing the type of precursor and tuning the growth temperature. Moreover, through simply mixing the precursors, a variety of 2D TMN alloys have been fabricated, further confirming the universality of this method.

## Characterizations of 2D TMNs and their alloys

By using the above RTF method, 7 types of mono-metal non-layered 2D TMNs (i.e., CrN, MnN, VN, FeN, CoN, h-NiN, r-NiN, WN, and ZrN) as well as 8 types of 2D TMN alloys (i.e., r-Co$_x$Ni$_y$N, h-Co$_{0.6}$Ni$_{0.4}$N, Co$_{0.3}$Fe$_{0.7}$N, Cr$_x$Fe$_y$N, Co$_{0.2}$Ni$_{0.1}$Fe$_{0.7}$N, Cr$_x$Fe$_y$Co$_z$N, Cr$_x$Fe$_y$Co$_z$Mn$_w$N, Cr$_x$Fe$_y$Co$_z$Ni$_w$N) are grown (Fig. 2a, b). Taking CoN as an example, the lateral size and thickness of the 2D flakes are determined by the growth temperature and duration of step 1 (Supplementary Fig. 10 and Supplementary Fig 11). The lateral size exhibits a volcano-like relationship with both increasing temperature and time, whereas the thickness generally increases. The largest CoN flake obtained thus far has a lateral size of 51 μm (Supplementary Fig. 12a), achieved at a growth temperature of 750 °C with a 5 min growth time, while the thinnest flakes (~1.03 nm, Supplementary Fig. 12b) are obtained at 600 °C with a 2 min growth time. We performed systematic studies and found that the morphology of the samples is decided by both their crystal symmetry and growth temperature. Most 2D TMNs and

alloys show hexagonal morphology, which is similar to their corresponding TMCls with six- or three-fold symmetry. Note that MnN, r-NiN, and r-Co$_x$Ni$_y$N show rectangular morphology, which is decided by the morphology of the template. For instance, as shown in Supplementary Fig. 13, 2D NiN exhibits a hexagonal morphology when synthesized at 650 °C, which corresponds to the hexagonal shape of the 2D template formed at that temperature. Conversely, when the step 1 temperature is increased to 800 °C, a rectangular shape 2D template forms, resulting in rectangular 2D NiN flakes. And this result may be attributed to their different composition of nickel chloride grown under different temperature. Taken together, we have grown 15 types of non-layered 2D TMNs and their alloys by using the RTF method.

We then investigated the crystal structure and composition of the 2D TMNs. Fig. 3a shows the crystal structure of Fe$_2$N (PDF#97-002-0390). The HRTEM image and selected area electron diffraction (SAED) pattern (Fig. 3b and its inset) reveals that the 2D Fe$_2$N has a lattice spacing of 0.138 nm, which is assigned to the (110) plane. Low magnification image and the corresponding energy-dispersive spectroscopy (EDS) mapping indicate that Fe and N atoms are uniformly distributed (Fig. 3c). The corresponding EDS result shows the characteristic peaks of Fe and N atoms (Fig. 3d) with an atomic ratio of ~2:1,

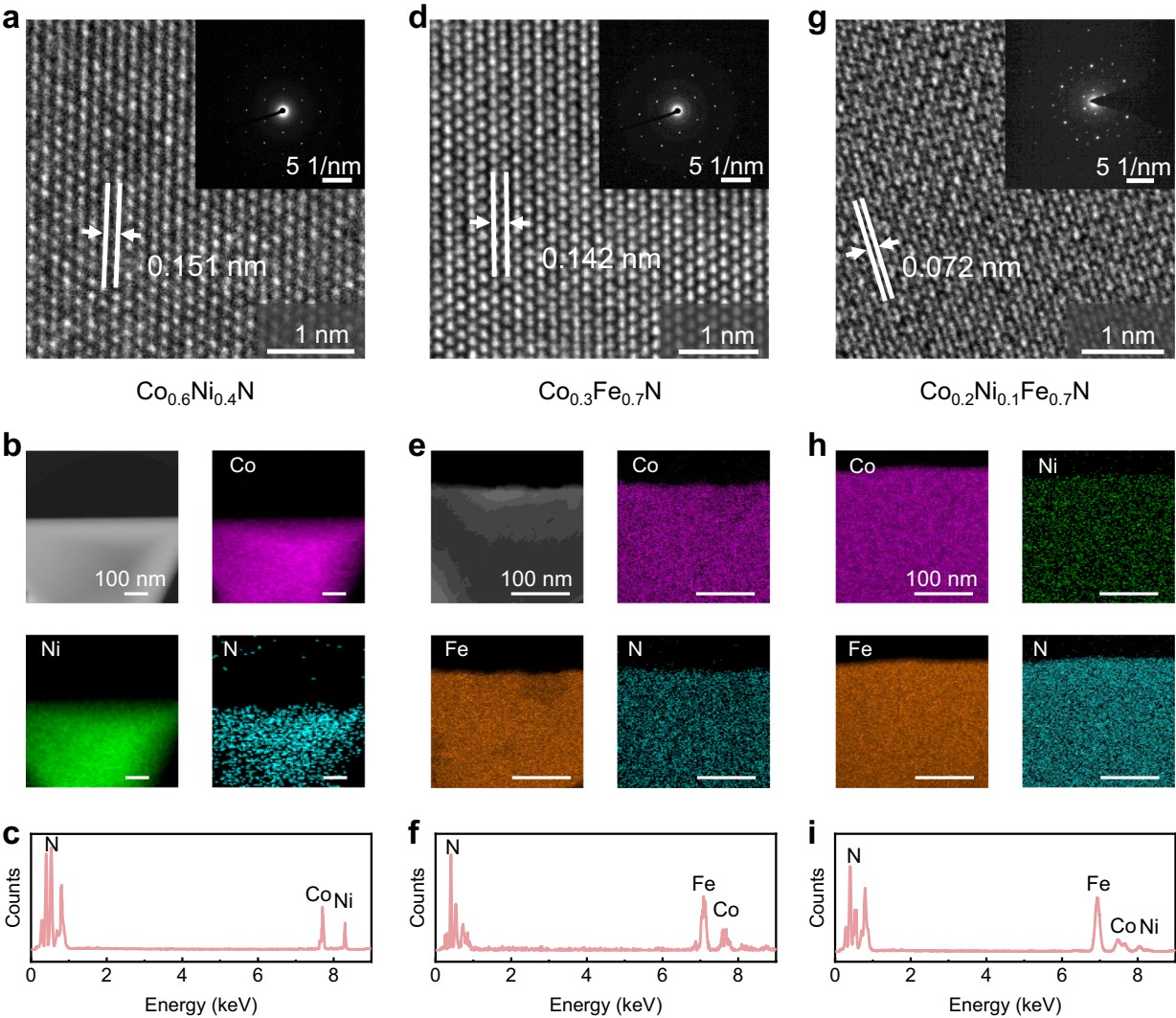

**Fig. 4 | Structural and composition characterizations of 2D TMN alloys.**
**a** HRTEM image and SAED pattern of the synthesized 2D $Co_{0.6}Ni_{0.4}N$ flake. **b** Low magnification TEM image, elemental EDS mapping of Co, Ni, and N atoms, and (**c**) EDS of the 2D $Co_{0.6}Ni_{0.4}N$ flake. **d** HRTEM image and SAED pattern of the synthesized 2D $Co_{0.3}Fe_{0.7}N$ flake. **e** Low magnification TEM image, elemental EDS mapping of Co, Fe and N atoms, and (**f**) EDS of the HRTEM image and SAED pattern of the synthesized 2D $Co_{0.3}Fe_{0.7}N$ flake. **g** HRTEM image and SAED pattern of the 2D $Co_{0.2}Ni_{0.1}Fe_{0.7}N$ flake. **h** Elemental EDS mapping of Co, Ni, Fe and N atoms and (**i**) EDS of the 2D $Co_{0.2}Ni_{0.1}Fe_{0.7}N$ flake.

suggesting the composition of $Fe_2N$. Similarly, we obtained HRTEM images of 2D VN and 2D MnN and their corresponding SAED patterns (Fig. 3f, j). The 2D VN shows a lattice spacing of 0.146 nm, and the 2D MnN shows a lattice spacing of 0.247 nm, which are consistent with their atomic structures (VN: PDF#97-016-9819; MnN: PDF#97-023-6828). The corresponding EDS mapping (Fig. 3g,k) and EDS results (Fig. 3h, l) show that all the elements are uniformly distributed and both the V: N and Mn: N atomic ratios are 1:1, which are in consistent with their corresponding crystal structures (Fig. 3e, i). The detailed HRTEM images and SAED patterns of the other 2D TMNs are shown in Supplementary Figs. 14–17, and the h-NiN and r-NiN belong to the same atomic structure (PDF#97-016-1755). These results suggest the high quality of the obtained 2D TMNs.

We also characterized the structure and composition of the 2D TMN alloys. HRTEM image and SAED pattern of the 2D $Co_{0.6}Ni_{0.4}N$ alloy are shown in Fig. 4a, revealing a lattice spacing of 0.151 nm, which is similar to the CoN ($d_{220}$ = 0.151 nm) and NiN ($d_{220}$ = 0.152 nm). Highly uniform elemental distribution is also confirmed by EDS mapping in Fig. 4b, and the corresponding EDS result (Fig. 4c) shows the characteristic peaks of Co, Ni and N atoms, with the atomic ratio of Co: Ni: N

is about 0.6: 0.4: 1. For $Co_{0.3}Fe_{0.7}N$, the HRTEM and SAED results in Fig. 4d reveal a lattice spacing of 0.142 nm, which is similar to the CoN ($d_{220}$ = 0.151 nm) and FeN ($d_{220}$ = 0.141 nm, PDF#97-023-6810). Interestingly, the FeN here belongs to the cubic structure rather than the previously fabricated 2D $Fe_2N$. The EDS mapping (Fig. 4e) exhibits the highly uniform elemental distribution of $Co_{0.3}Fe_{0.7}N$ alloy. The corresponding EDS result in Fig. 4f shows the characteristic peaks of Fe, Ni, and N atoms, with the atomic ratio of Fe: Co: N is about 0.7: 0.3: 1. Moreover, the HRTEM image of 2D $Co_{0.2}Ni_{0.1}Fe_{0.7}N$ and its corresponding SAED pattern (Fig. 4g) show a lattice spacing of 0.072 nm, which is almost half of NiN ($d_{220}$ = 0.152 nm), CoN ($d_{220}$ = 0.151 nm), and FeN ($d_{220}$ = 0.141 nm, PDF#97-023-6810). This should be attributed to the lattice distortion caused by the different lattice constants of the three materials. Additionally, the corresponding EDS mapping (Fig. 4h) and EDS result (Fig. 4i) demonstrate the uniformly distribution of Ni, Co and Fe in the flakes with an approximate atomic ratio of Co: Ni: Fe: N = 0.2: 0.1: 0.7: 1. The detailed HRTEM image and EDS mapping of 2D r-$Co_xNi_yN$ are shown in Supplementary Fig. 18. The above results suggest that the RTF synthesized 2D TMN alloys exhibit high crystallinity and uniformity.

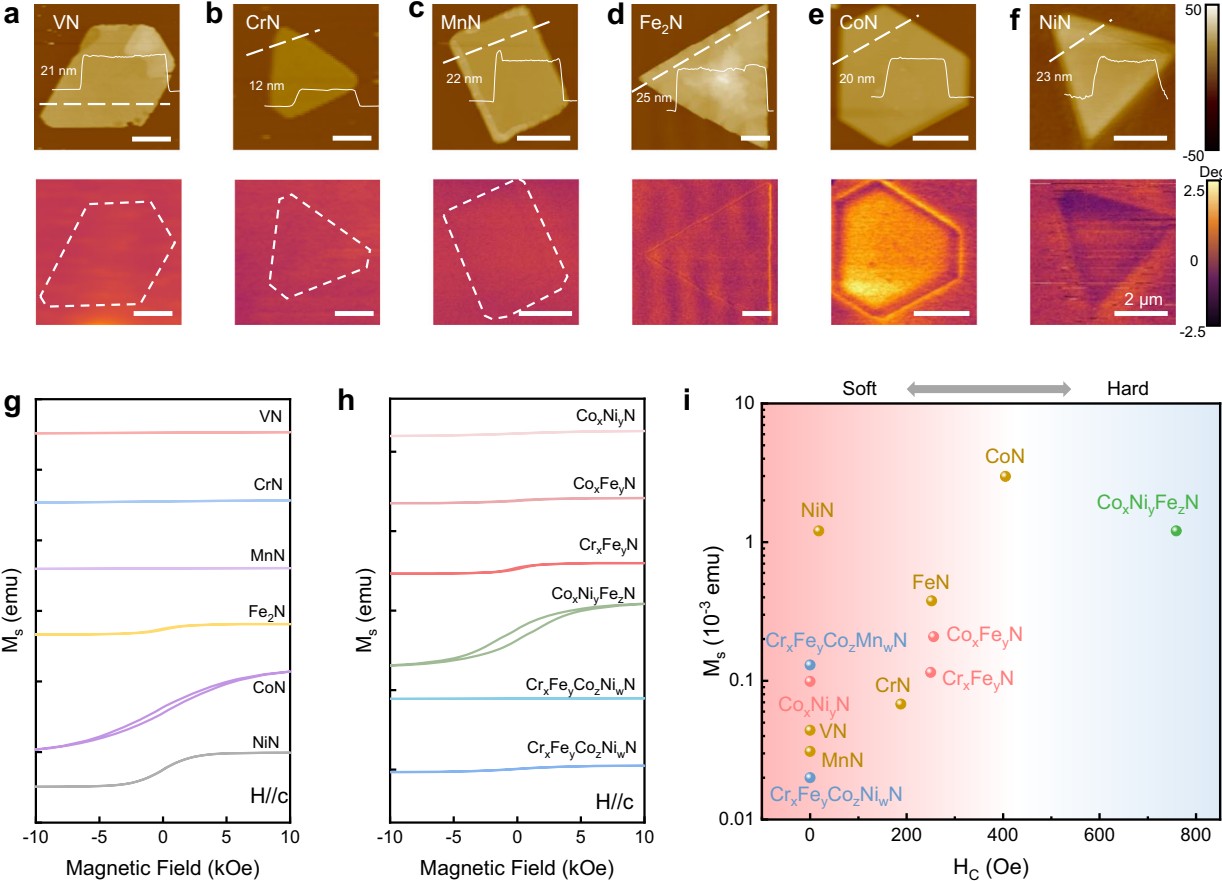

**Fig. 5 | Magnetism of 2D TMNs and their alloys.** Atomic force microscopy (AFM) and magnetic force microscopy (MFM) images of individual (**a**)2D VN, (**b**) 2D CrN, (**c**) 2D MnN, (**d**) 2D Fe$_2$N, (**e**) 2D CoN, and (**f**) 2D NiN flakes measured at R.T. The scale bar is 2 μm. **g** Magnetization versus magnetic field (M-H) loops of the 2D TMNs and (**h**) their alloys with the magnetic field applied out of plane. The results are measured by superconducting quantum interference device (SQUID) at 10 K.

The effect of mica substrate has been eliminated (Supplementary Fig. 19). **i** Statistics of coercive force (H$_c$) and saturation magnetization (M$_s$) of the 2D TMNs and their alloys in (**g**) and (**h**). The materials with one, two, three and four types of metal elements are highlighted in yellow, pink, green and blue, respectively. And increasing H$_c$ drives the transition from soft to hard magnetic materials.

## Magnetic properties of 2D TMNs and their alloys

Finally, we investigated the magnetic properties of 2D TMNs and their alloys by magnetic force microscopy (MFM) and superconducting quantum interference device (SQUID). Fig. 5a-f show the MFM phase contrasts of the 2D TMNs (i.e., VN, CrN, MnN, Fe$_2$N, CoN, and NiN) at R.T. The results show that CoN has the largest saturation magnetization among these materials, which is consistent with their M-H loops measured by SQUID (Fig. 5g). The M-H loops of 2D TMN alloys were also characterized and summarized in Fig. 5h. In detail, the 2D Co$_x$Fe$_y$N alloy shows an antiferromagnetic behavior, with lower saturation magnetization (M$_s$) and coercive force (H$_c$) than both 2D CoN and 2D Fe$_2$N. While 2D Cr$_x$Fe$_y$N alloy exhibits a ferromagnetic behavior (2D Fe$_2$N: antiferromagnetic, 2D CrN: ferromagnetic) and moderate magnetism (M$_s$ = 0.208 emu, H$_c$ = 250 Oe) compared with 2D CrN (M$_s$ = 0.068 emu, H$_c$ = 188 Oe) and 2D Fe$_2$N (M$_s$ = 0.377 emu, H$_c$ = 252 Oe). The magnetic properties of all the 2D TMNs and their alloys are summarized in Fig. 5i and Supplementary Table 1. Taken together, we can either enhance or weaken the magnetic response of 2D TMNs by selecting appropriate metals or alloying, thereby expanding the performance range of 2D magnets to meet diverse application requirements.

## Discussion

In this work, we developed a RTF CVD method to convert 2D TMCls and obtained fifteen types of non-layered 2D TMNs and their alloys.

The key of the method is using metastable TMCls as transient templates, with features of their 2D layered structure and low conversion energy barrier to TMNs. Combined HRTEM images, SAED patterns and EDS spectra results reveal the single-crystalline structure and composition uniformity of these 2D materials. Magnetism characterizations show rich and tunable magnetic behaviors of 2D TMN by alloying. The general growth method for non-layered materials could enrich the 2D material family with exotic properties and applications.

## Methods

### Synthesis of 2D TMNs and their alloys

The 2D TMNs and their alloys were fabricated using the RTF method. The typical fabrication process was carried out in a movable furnace, as shown in Supplementary Fig. 6. The precursors were placed in a quartz boat located in the upstream region of the quartz tube (Field I), and the mica substrates were positioned in the downstream region (Field II). Argon (100 sccm) was used as the carrier gas throughout the process. The growth involved two distinct steps: Step 1 involved the deposition of the 2D TMCl templates. In this step, Field I (precursor zone) was moved to the heating center and heated to the target temperature (600-800 °C) from room temperature in 30 min and maintained for 2 min. During this step, Field II (substrate zone), situated approximately 10 cm away from the heating center, remained at a lower temperature (100-200 °C). At the end of Step 1, the furnace was swiftly translated along the rail to position Field II (substrate zone) at the

heating center, initiating Step 2 (the conversion step). Simultaneously with this movement, a mixture of $NH_3$ and $H_2$ was introduced and maintained for 10 min. Consequently, the temperature of Field II was rapidly increased to 600-800 °C and held for 5 min, while Field I was cooled down to approximately 100-200 °C. Finally, the furnace was cooled to room temperature under an Ar atmosphere.

Traditional CVD growth of 2D TMN: the precursor was placed in the upstream section of a one-zone furnace (Field I), while the substrate was positioned at the furnace center (Field II, the heating center). In step 1, the precursor was heated to 300-400 °C (above its sublimation temperature) and carried by 100 sccm Ar to deposit onto the substrate. In step 2, the substrate was heated to 600-800 °C under a flow of $H_2$ and $NH_3$ to facilitate the nitridation of the deposited TMCl. The corresponding temperature profile is illustrated in Supplementary Fig. 9.

Transfer of 2D TMNs and their alloys: The 2D flakes grown on the mica substrates were transferred onto the TEM grid by a common method using Polystyrene (PS) as a sacrificial transfer film. PS was spincoated on the sample at 2000 rpm for 60 s. The sample was then heated on a hot plate at 90 °C for 5 min. After baking, the sample was loaded in deionized water and then the peeled-off samples were transferred onto TEM grids. Finally, the PS was removed with methylbenzene.

Material characterizations and magnetic measurements: The morphology of the samples was observed by an optical microscope (Carl Zeiss Microscopy, Germany). The microstructure of the samples was investigated by high-resolution TEM at an acceleration voltage of 300 kV (FEI Tecnai F30, USA). SQUID (MPMS3, Quantum Design, USA) was used to measure the M-H hysteresis loops and M-T curves of the samples. AFM (Cypher ES, Asylum Research, USA) was used to measure the thickness of the samples. MFM (Cypher ES, Asylum Research, USA) was used to measure the magnetic properties of the nanoflakes.

Calculations of temperature-dependent Gibbs formation energy of compounds: Enthalpy data of compounds in this work were extracted from Lange's handbook of chemistry[38] (CrN, $CrCl_3$, $Cr_2O_3$, $FeCl_3$, $Fe_2O_3$, FeS, $MnCl_2$, $MnO_2$, $NiCl_2$, NiO, NiS, CoO), FactSage[39] (CrS, $CoCl_2$, CoS, $Fe_2N$, MnS, $VCl_3$, $V_2O_5$, $VS_2$), and DFT calculation (CoN, NiN, MnN, VN). All the compound data were extracted only at 298 K. The chemical potentials of the elements were obtained with the minimum Gibbs energy at a given temperature.

## Data availability
The Source Data underlying the figures of this study are available with the paper. All raw data generated during the current study are available from the corresponding authors upon request. Source data are provided with this paper.

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

## Acknowledgements

This work was supported by the National Science Foundation of China for Distinguished Young Scholars (52125309), the National Key R&D Program of China (2022YFA1204301), and the National Natural Science Foundation of China (52188101, 62404124 and 52473308), Innovation Team Project of Department of Education of Guangdong Province (2023KCXTD051), the Shenzhen Basic Research Project (JCYJ20230807111619039 and JCYJ20220818101014029), and Natural Science Foundation of Guang-dong Province of China (2023A1515011752), Shenzhen Science Technol-ogy Program (ZDSYS20230626091100001), and Tsinghua Shenzhen International Graduate School-Shenzhen Pengrui Young Faculty Program of Shenzhen Pengrui Foundation (No. SZPR2023002), the Shenzhen HanHua TM Technology Co., LTD (20249660086), and the Jiangsu Han-Hua TM Technology Co., LTD (20249660085). This work made use of the TEM facilities at the Institute of Materials Research, Tsinghua Shenzhen International Graduate School (Tsinghua SIGS). The authors would like to thank Units Technology Co.,Ltd (www.units-tech.com.cn) for offering the high-temperature in situ visualization setup.

## Author contributions

B.L. conceived the study and supervised the project. L.H. designed and performed experiments, analyzed data and prepared the manuscript. J.W. performed data collection and analysis. Z.C. carried out the theo-retical calculations. R.L. assisted in the synthesis. S.L., Y.Z., Z.Y. Z. and J.L. assisted in the characterizations. B.L., J.W. and C.Z. helped with article revisions. All the authors discussed the results and commented on the manuscript.

## Competing interests

The authors declare no competing interests.
