## [Transparent Peer Review file · Nature Communications]

Growth of non-layered 2D transition metal nitrides enabled by transient chloride templates

Corresponding Author: Professor Bilu Liu

Version 0:

Reviewer comments:

Reviewer #1

(Remarks to the Author)

Non-layered 2D materials have attracted much attention recently and is a new research frontier in the 2D materials field. Among all non-layered 2D materials, transition metal nitrides TMNs are particularly interesting but is rather difficult to grow into 2D thin forms due to their isotropic covalent/ionic bonding, which is a well-known challenge in the field. In this paper, He et al. have established a very innovative method, by using metal chloride as a “Transient Template”, to grow a large variety of metal nitrides and their alloys into large and thin 2D forms. This seems to be the first report about universal synthesis of such large amount of different kinds of 2D TMNs. The authors have performed systematical investigations regarding material growth mechanism, material characterization, structural and chemical analysis, as well as magnetism properties. Overall, the quality of research is high and the conclusions are supported by their experiments. This work has solved a long-standing problem in the field, and the referee recommends its publication in Nature Communications after minor revisions to further strengthen the work, noted below.

- 1). What's the largest size of 2D TMN flakes you can grow? What are the factors that limit the sizes of the 2D TMNs? Please show some experimental results or add discussions.
- 2). The figure 1 reflects the design concept of the new growth strategy which is nice. There seems a display typo, i.e., “metastable layered TMCl template?” Please check. In addition, what are the molecules in Figure 1c? It is not very clear to see, nor were explained or illustrated in the figure. Please make this point clear. Also, what's the temperature profile for traditional CVD? Please write it clearly.
- 3). Different 2D TMN materials seem have different shape or morphology, is that correct? Is it purely determined by the symmetry of the crystals, or also been affected by the growth parameters like growth temperature? Please discuss this point.
- 4). For the methods part, please describe more details about the growth furnace. Is it a one-zone or three-zone furnace? What's the position/temperature where you put the precursors? What's the detailed shape of the container? It would be helpful to add these details, preferably a photo of the set-up, so that others can use the growth recipe for these kinds of 2D TMN synthesis.
- 5). The English writing should be further polished. For example, in Abstract, “15 types of 2D TMNs and their alloys...” can be changed to “Fifteen types of 2D TMNs and their alloys...” for clarity. Figure 2 caption, “Number of Metal Elements”, the letter M and E may not be capitalized. Please keep consistent with other places throughout the manuscript. The authors should polish these details before publication.

Reviewer #2

(Remarks to the Author)

This work reports a novel and universal synthesis method, reverse-thermal-field (RTF) CVD strategy, for growing a wide range of non-layered 2D TMNs and their alloys. The key innovation is the use of metastable, layered TMCl as transient templates, which are first grown at a low-temperature zone and then rapidly converted into TMNs by quickly moving to a high-temperature zone in the presence of ammonia. The authors successfully demonstrate the growth of 15 different 2D TMNs (7 mono-metal nitrides and 8 alloys) and show that their magnetic properties (e.g., saturation magnetization, coercive

field) can be effectively tuned through composition. In general, this work is interesting and has good novelty. I can recommend the publication after addressing the following concerns:

1. The claimed mechanism is the “solid-to-solid” conversion of a 2D TMCl template. However, it is important to provide direct evidence that the layered structure of the initial TMCl flake is converted to TMN during the rapid heating step and acts as a true template, other than the direct deposition of TMN under the environment of precursor gas, NH₃, and H₂. Therefore, providing an optical image of TMCl flakes after step 1 (30 min heating + 2 waiting time) and before step 2 is necessary. Alternatively, it is also acceptable to perform the TEM characterization of the intermediate flakes during step 2, that should contain both Cl and N atoms.
2. It is hard to imagine how the TMCl_s with six- or three-fold symmetry transform into the rectangular-shaped TMNs. If it is a solid-to-solid transition, the metal atoms need to move freely on the mica substrate, and an intermediate shape between the triangle (or hexagon) and rectangle can be captured if step 2 is stopped at around 2.5 min.
3. It would be beneficial to include references for the sentence in lines 66-69.
4. In Fig. 5a-f, the thicknesses are not directly displayed; line cuts are necessary.

Reviewer #3

(Remarks to the Author)

The authors report a universal reverse-thermal-field CVD strategy for synthesizing two-dimensional transition metal nitrides (TMNs). In this approach, metastable metal chloride crystal templates are first deposited on mica substrates and then converted to TMNs by high-temperature nitridation, before the chloride templates are lost. Using this method, the authors demonstrate the growth of VN, CrN, MnN, Fe₂N, CoN, NiN, and their alloys. Structural and compositional characterizations (TEM, EDS, AFM) are presented, while magnetic properties are probed by MFM and SQUID. Overall, the technique represents a potentially viable route toward 2D TMNs and is therefore an important contribution to the 2D materials community. However, additional details and clarifications are needed:

1. A more detailed parametric study of the different materials should be conducted. For example, how was the temperature for Field 1 and 2 selected? Was any optimization performed in selecting these values? Can the crystal quality/defect density of the as-grown TMN be quantified? Such as using Raman peak width/position, and corroborated with XRD/XPS if feasible.
2. Can the CVD parameters, such as precursor loading or growth duration, be used to systematically control layer thickness, nucleation density, and flake size? Since the authors describe these TMNs as 2D materials, please report the thickness statistics, which should typically be in the range of a few nm. Further optimization would be necessary to probe genuine 2D characteristics rather than bulk-like properties.
3. Figure S2 shows the rapid loss of metastable CrCl₃ crystal within 40 s. For the RTF-CVD process, please provide the time-temperature profile. How rapid is the heating/cooling transition from 100–200 °C up to 600–800 °C? This is important for researchers to replicate the method.
4. For the alloyed TMNs, are the metal elements randomly incorporated in a single phase, or is there any short-range ordering/phase segregation? Can the elemental composition be tuned by adjusting precursor ratios?
5. The AFM and MFM images of the alloyed TMN, along with the corresponding thickness measurements, should be included in Figure 5. Does the magnetic property change with thickness?

Version 1:

Reviewer comments:

Reviewer #1

(Remarks to the Author)

The authors have revised the manuscript well, I recommend to accept it for publication.

Reviewer #2

(Remarks to the Author)

All of my comments have been thoroughly addressed. Therefore, I recommend the publication.

Reviewer #3

(Remarks to the Author)

The authors have addressed all the raised concerns. I find the revised manuscript suitable for publication and recommend acceptance.

List of changes (blue in the revised manuscript and SI):

- (1) Some sentences have been added or revised on Pages 2, 4, 7, 10, 18, 19 and 20 in the revised manuscript.
- (2) Figure 1b, Figure 2 and Figures 5a-5f have been updated in the revised manuscript according to the suggestions of reviewers.
- (3) The in-situ observation of deposited TMCl evolution under high temperature has been added as Figure S3 in the revised SI.
- (4) A photo of the setup used to achieve RTF process has been added as Figure S6 in the revised SI.
- (5) Temperature profile of the growth process of RTF method has been added as Figure S8 in the revised SI.
- (6) Temperature profile of the traditional CVD method has been added as Figure S9 in the revised SI
- (7) The relationship between morphology of CoN and growth temperature has been added as Figure S10 in the revised SI.
- (8) The relationship between morphology of CoN and growth time has been added as Figure S11 in the revised SI.
- (9) OM and AFM images of 2D CoN have been added as Figure S12 in the revised SI.
- (10) OM images of 2D nickel chloride templates with different morphologies have been added as Figure S13 in the revised SI.
- (11) Typos have been corrected in the revised manuscript.
- (12) Two new references have been added in the revised manuscript.

Response to Reviewer #1

Comment. Non-layered 2D materials have attracted much attention recently and is a new research frontier in the 2D materials field. Among all non-layered 2D materials, transition metal nitrides TMNs are particularly interesting but is rather difficult to grow into 2D thin forms due to their isotropic covalent/ionic bonding, which is a well-known challenge in the field. In this paper, He et al. have established a very innovative method, by using metal chloride as a “Transient Template”, to grow a large variety of metal nitrides and their alloys into large and thin 2D forms. This seems to be the first report about universal synthesis of such large amounts of different kinds of 2D TMNs. The authors have performed systematical investigations regarding material growth mechanism, material characterization, structural and chemical analysis, as well as magnetism properties. Overall, the quality of research is high and the conclusions are supported by their experiments. This work has solved a long-standing problem in the field, and the referee recommends its publication in Nature Communications after minor revisions to further strengthen the work, noted below.

Response. Thank you very much for your positive and encouraging assessment of our work. We sincerely appreciate the reviewer by writing that our work is “very innovative”, “systematical investigations” and pointing out “to be the first report about universal synthesis of such large amounts of different kinds of 2D TMNs”.

Comment 1. What’s the largest size of 2D TMN flakes you can grow? What are the factors that limit the sizes of the 2D TMNs? Please show some experimental results or add discussions.

Response 1. We thank the reviewer for these insightful questions. (1) Regarding the largest size, taking CoN as an example, the largest size is **51 μm** . (2) Regarding the factors that affect and limit the sizes of the 2D TMN, our results reveal that the lateral size of the 2D TMNs is primarily determined by the **growth temperature** and **growth time** of step 1 (the template formation step), as detailed in Figures R1 and R2. Both parameters exhibit a volcano-shaped relationship with the lateral size of the resulting 2D CoN flakes. This trend can be attributed to the nucleation density of the transition metal chloride (TMCl) templates: at lower nucleation densities, increasing the temperature or time promotes lateral growth, leading to larger flakes. However, beyond an optimal point, excessively high nucleation density restricts the available

space for lateral expansion, thereby limiting the final flake size. Under the optimized conditions (a growth temperature of 750 °C and a growth time of 5 minutes), we have achieved 2D CoN flakes with an average lateral size of several tens of micrometers, with the largest flake reaching 51 μm (Figure R3).

Figure R1. The relationship between morphology of CoN and growth temperature. (a) OM images of CoN with increased temperature from 600 °C to 800 °C, and all the growth time were 5 min. (b) The thicknesses of CoN under different growth temperatures. The thickness was obtained by AFM measurements of 249 flakes. (c) The lateral sizes of CoN under different growth temperature. The lateral size was obtained by OM characterization.

Figure R2. The relationship between morphology of CoN and growth time. (a) OM images of CoN with increased growth time from 2 min to 20 min, and all the growth temperatures were 650 °C. (b) The thicknesses of CoN with different growth time. The thickness was obtained by AFM measurements of 293 flakes. (c) The lateral sizes of CoN with different growth time. The lateral size was obtained by OM characterization.

Figure R3. The OM image of 2D CoN with the lateral size of 51 μm .

Changes to the manuscript.

- Figure R1, Figure R2 and Figure R3 have been added as **Figure S10**, **Figure S11** and **Figure S12a** in the revised SI.
- Several sentences have been added on **Page 10**. “Taking CoN as an example, the lateral size and thickness of the 2D flakes are determined by the growth temperature and duration of step 1 (Figure S10 and S11). The lateral size exhibits a volcano-like relationship with both increasing temperature and time, whereas the thickness generally increases. The largest CoN flake obtained thus far has a lateral size of 51 μm (Figure S12a), achieved at a growth temperature of 750 $^{\circ}\text{C}$ with a 5-minute growth time, while the thinnest flakes (~ 1.03 nm, Figure S12b) are obtained at 600 $^{\circ}\text{C}$ with a 2-minute growth time.”

Comment 2. The figure 1 reflects the design concept of the new growth strategy which is nice. There seems a display typo, i.e., “metastable layered TMCl template?” Please check. In addition, what are the molecules in Figure 1c? It is not very clear to see, nor were explained or illustrated in the figure. Please make this point clear. Also, what’s the temperature profile for traditional CVD? Please write it clearly.

Response 2. We sincerely thank the reviewer for pointing out these oversights. We have carefully corrected the typo in the figure. Furthermore, we have revised Figure 1 (now Figure R4 in the response) to provide a clearer illustration of the molecular species involved and to explicitly label the temperature profiles. Specifically, the molecules depicted in Figure 1c

represent the sublimated metal chloride species (e.g., MCl_x). Regarding the traditional CVD process, its temperature profile is now provided in the new Figure S8. In a standard one-zone CVD setup, the precursor (located upstream, Field I) is typically heated to 300-400 °C (above its sublimation point) and transported by Ar carrier gas to deposit on the substrate (located at the center, Field II). Subsequently, in step 2, the substrate region is heated to 600-800 °C under a mixture of H_2 and NH_3 for nitridation. However, due to the instability of the 2D TMCl templates at high temperatures, they often sublime before the nitridation temperature is reached in this configuration, leading to unsuccessful growth.

Figure R4. Design and growth process of the reverse-thermal-field (RTF) CVD method.

(a) Calculations of the Gibbs energy barriers of the solid-to-solid conversion from transition metal chlorides, oxides, and sulfides to the TMNs counterparts at 1000 K. The positive energy

means the conversion is theoretically unfavorable, while the negative energy means it is theoretically favorable. (b) Schematic of the metastable 2D TMCl template. (c) Scheme to illustrate the temperature program and growth result of the traditional CVD method. The nitridation process starts when the temperature reaches the thermodynamically supported conversion temperature. And the TMCl is sublimated as MCl_x molecules before the nitridation process. (d) Scheme to illustrate the temperature program and growth result of the RTF CVD method. The dashed and solid lines represent the temperature of the precursor and substrate, respectively. The nitridation process starts when the thermal field is reversed. The detailed process is shown in the Experimental Section and Supporting Information.

Figure R5. Photo of the setup used to achieve RTF process.

Figure R6. Temperature profile of the traditional CVD method. (a) Temperature profile of the precursor during growth, and the precursor is located at Field I. (b) Temperature profile of the substrate during growth, and the substrate is located at Field II.

Changes to the manuscript.

- Figure R4 has been reproduced as **Figure 1** in the revised manuscript to replace the original Figure 1.
- Figure R5 and Figure R6 have been added as **Figure S6** and **Figure S9** in the revised SI
- Several sentences have been added on **Pages 19 and 20**. “*Traditional CVD growth of 2D TMN*: the precursor was placed in the upstream section of a one-zone furnace (Field I), while the substrate was positioned at the furnace center (Field II, the heating center). In step 1, the precursor was heated to 300-400 °C (above its sublimation temperature) and carried by 100 sccm Ar to deposit onto the substrate. In step 2, the substrate was heated to

600-800 °C under a flow of H₂ and NH₃ to facilitate the nitridation of the deposited TMCl.
The corresponding temperature profile is illustrated in Figure S9.”

Comment 3. Different 2D TMN materials seem have different shape or morphology, is that correct? Is it purely determined by the symmetry of the crystals, or also been affected by the growth parameters like growth temperature? Please discuss this point.

Response 3. Thank you for raising this important point regarding morphology control. (1) Different 2D TMNs indeed show different shapes under different growth conditions. (2) The final morphology of the 2D TMNs is indeed directly inherited from the **morphology of the intermediate 2D TMCl template** formed during step 1. The growth parameters, particularly the temperature in step 1, critically influence the shape of this initial template. For instance, as shown in Figure R7, 2D NiN exhibits a hexagonal morphology when synthesized at 650 °C, which corresponds to the hexagonal shape of the 2D template formed at that temperature (Figure R7a). Conversely, when the step 1 temperature is increased to 800 °C, a rectangular shape 2D template forms (Figure R7b), resulting in rectangular 2D NiN flakes. Therefore, the morphology is predominantly determined by the growth-parameter-dependent morphology of the transient TMCl template.

Figure R7. OM images of 2D nickel chloride templates obtained at different growth temperatures. (a) OM image of a triangular/hexagonal shape 2D template when the growth temperature is 650 °C. (b) OM image of a rectangular shape 2D template when the growth temperature is 800 °C.

Changes to the manuscript.

- Figure R7 has been added as **Figure S13** in the revised SI
- Several sentences have been added on **Page 10**. “Note that MnN, r-NiN, and r-Co_xNi_yN

show rectangular morphology, which is decided by the morphology of the template. For instance, as shown in Figure S13, 2D NiN exhibits a hexagonal morphology when synthesized at 650 °C, which corresponds to the hexagonal shape of the 2D template formed at that temperature. Conversely, when the step 1 temperature is increased to 800 °C, a rectangular shape 2D template forms, resulting in rectangular 2D NiN flakes.”

Comment 4. For the methods part, please describe more details about the growth furnace. Is it a one-zone or three-zone furnace? What’s the position/temperature where you put the precursors? What’s the detailed shape of the container? It would be helpful to add these details, preferably a photo of the set-up, so that others can use the growth recipe for these kinds of 2D TMN synthesis.

Response 4. We appreciate the reviewer’s suggestion to enhance the reproducibility of our method. As requested, we have provided a photograph of our setup (Figure R8) and the detailed temperature profile for the RTF process (Figure R9). The growth was performed in a modified one-zone tube furnace equipped with a sliding rail system, which allows the furnace to be physically translated between Field I (precursor location) and Field II (substrate location). The precursors were contained in a standard quartz boat. We have thoroughly revised the “Experimental Section” in the manuscript to incorporate these crucial experimental details, ensuring the growth recipe is clear for other researchers.

Figure R8. Photo of the setup used to achieve RTF process.

Figure R9. Temperature profile of the growth process of RTF method. (a) Temperature profile of Field I, where the precursor is located. (b) Temperature profile of Field II, where the substrate is located.

Changes to the manuscript.

- Figure R8 and Figure R9 has been added as **Figure S6** and **Figure S8** in the revised SI.
- Several sentences have been rewritten on **Page 7** in the revised manuscript. “Figures 1d and S6-S8 illustrate the set-up of the furnace and the growth process of RTF method and traditional CVD method (see details in Experimental Section and Supporting Information).”
- Several sentences have been added on **Page 20** in the revised manuscript. “*Synthesis of 2D TMNs and their alloys*: the 2D TMNs and their alloys were fabricated using the RTF method. The typical fabrication process was carried out in a movable furnace, as shown in Figure S6. The precursors were placed in a quartz boat located in the upstream region of the quartz tube (Field I), and the mica substrates were positioned in the downstream region (Field II). Argon (100 sccm) was used as the carrier gas throughout the process. The growth involved two distinct steps: Step 1 involved the deposition of the 2D TMCl templates. In this step, Field I (precursor zone) was moved to the heating center and heated to the target temperature (600-800 °C) from room temperature in 30 minutes and maintained for 2 minutes. During this step, Field II (substrate zone), situated approximately 10 cm away from the heating center, remained at a lower temperature (100-200 °C). At the end of Step 1, the furnace was swiftly translated along the rail to position Field II (substrate zone) at the heating center, initiating Step 2 (the conversion step). Simultaneously with this movement, a mixture of NH₃ and H₂ was introduced and maintained for 10 minutes. Consequently, the temperature of Field II rapidly increased to 600-800 °C and was held for 5 minutes, while Field I cooled down to approximately 100-200 °C. Finally, the furnace was cooled to room temperature under an Ar atmosphere.”

Comment 5. The English writing should be further polished. For example, in Abstract, “15 types of 2D TMNs and their alloys...” can be changed to “Fifteen types of 2D TMNs and their alloys...” for clarity. Figure 2 caption, “Number of Metal Elements”, the letter M and E may not be capitalized. Please keep consistent with other places throughout the manuscript. The authors should polish these details before publication.

Response 5. We sincerely thank the reviewer for this meticulous feedback on language and formatting. We have carefully polished the English throughout the entire manuscript.

Specifically, the instance in the abstract has been changed to “Fifteen types...”. The capitalization in the Figure 2 caption has been corrected to “Number of metal elements” for consistency. We have conducted a thorough proofreading to ensure uniformity in terminology, capitalization, and overall language quality.

Figure R10. Universal growth of 2D TMNs and their alloys by the RTF method. (a) The periodic table of elements shows the transition metal elements involved in this work. (b) Optical microscopy images of the 15 types of 2D TMNs and alloys. All the scale bars are 5 μm . The materials have one, two, three, and four types of metal elements are highlighted in yellow, pink, green, and blue, respectively.

Changes to the manuscript.

- Figure R10 has been added as **Figure 2** in the revised manuscript to replace the original Figure 2.
- The sentence on **Page 2** has been rewritten. “Fifteen types of 2D TMNs and their alloys

were synthesized, demonstrating the versatility of this method.”

- The sentence on **Page 18** has been rewritten. “In this work, we developed a reverse-thermal field CVD method to convert 2D TMCl and obtained fifteen types of non-layered 2D TMNs and their alloys.”

Response to Reviewer #2

Comment. This work reports a novel and universal synthesis method, reverse-thermal-field (RTF) CVD strategy, for growing a wide range of non-layered 2D TMNs and their alloys. The key innovation is the use of metastable, layered TMCl as transient templates, which are first grown at a low-temperature zone and then rapidly converted into TMNs by quickly moving to a high-temperature zone in the presence of ammonia. The authors successfully demonstrate the growth of 15 different 2D TMNs (7 mono-metal nitrides and 8 alloys) and show that their magnetic properties (e.g., saturation magnetization, coercive field) can be effectively tuned through composition. In general, this work is interesting and has good novelty. I can recommend the publication after addressing the following concerns:

Response. Thanks very much to the reviewer for your careful reading and positive recommendations. We appreciate the reviewer for the comments that “novel and universal synthesis method” and “interesting and has good novelty”. And according to the suggestions, we have made extensive revisions to improve the quality of our manuscript.

Comment 1. The claimed mechanism is the “solid-to-solid” conversion of a 2D TMCl template. However, it is important to provide direct evidence that the layered structure of the initial TMCl flake is converted to TMN during the rapid heating step and acts as a true template, other than the direct deposition of TMN under the environment of precursor gas, NH_3 , and H_2 . Therefore, providing an optical image of TMCl flakes after step 1 (30 min heating + 2 waiting time) and before step 2 is necessary. Alternatively, it is also acceptable to perform the TEM characterization of the intermediate flakes during step 2, that should contain both Cl and N atoms.

Response 1. We sincerely thank the reviewer for this valuable suggestion, which is crucial for validating the “solid-to-solid” conversion mechanism. Regarding TEM characterization, we have tried multiple times to transfer the “transient” TMCl templates to TEM grids for TEM characterization, but failed. We found that these transient templates are not stable and disappear rapidly at ambient conditions (< 1 min).

To directly observe this process, we followed your suggestions and conducted new experiments using a visualization CVD system. Such a setup allows us in-situ monitor of the through a viewport by optical microscopy. In this setup (Figure R12a), the precursor is located in zone A and the substrate in zone B and Figure R11 illustrates the temperature profiles of zone A and zone B. As shown in Figure R12b, the TMCl template is first deposited on the substrate when zone A is heated above the precursor sublimation temperature. Subsequently, we captured the intermediate state during the transition to step 2. When NH_3 is not introduced (Figure R12c, Figure R13), the TMCl template sublimates away quickly due to the high temperature, leaving no solid product. In contrast, when NH_3 is introduced (Figure R12d), the TMCl template is converted in-situ into TMN, which retains its structural integrity. This direct observation provides compelling evidence that the **conversion is indeed a solid-to-solid process, rather than a direct deposition from the gas phase.**

Figure R11. The growth process of the visualization two-zone CVD. (a) The temperature profile of zone A, where the precursor located. (b) The temperature profile of zone B, where the substrate located.

Figure R12. In-situ observation of TMCl evolution by visualization CVD. (a) In-situ visualization two-zone CVD technique. (b) The OM images captured at zone B during step 1, demonstrating the deposition of CoCl_2 . (c) The OM images captured at zone B during the intermediate state and step 2 without the introduction of NH_3 , showing sublimation. (d) The OM images captured at zone B during the intermediate state and step 2 with the introduction of NH_3 , showing in-situ conversion to CoN . (e) The high-resolution OM image of the growth results without the introduction of NH_3 . (f) The high-resolution OM image of the growth results with the introduction of NH_3 .

Changes to the manuscript.

- A new figure reproduced from Figure R12 has been added as **Figure S3** in the revised SI
- Several sentences have been added on **Page 6**. “Indeed, Figure S2 shows the optical microscopy (OM) images of a 2D TMCl flake exposed in air for 10 s, 20 s, 30 s, 40 s, and 50 s, and Figure S3 shows the OM images of deposited TMCl heated at 650 °C in an Ar atmosphere. The quick fading of the flake demonstrates its poor stability.”

Comment 2. It is hard to imagine how the TMCl_s with six- or three-fold symmetry transform into the rectangular-shaped TMNs. If it is a solid-to-solid transition, the metal atoms need to move freely on the mica substrate, and an intermediate shape between the triangle (or hexagon) and rectangle can be captured if step 2 is stopped at around 2.5 min.

Response 2. We thank the reviewer for this insightful question. (1) The key clarification is that the final morphology of the TMN is not transformed from a different initial shape, but is directly inherited from the specific morphology of the transient TMCl template formed during step 1. (2) Taking nickel chloride as an example, as shown in Figure R13, at a step 1 temperature of 650 °C, a triangular shape template forms, leading to triangular h-NiN. Conversely, at a higher step 1 temperature of 800 °C, a rectangular shape template is obtained, which subsequently converts into rectangular r-NiN. Therefore, the rectangular morphology originates from a rectangular precursor template, not from a restructuring of a triangular one.

Figure R13. OM images of 2D nickel chloride templates obtained at different growth temperatures. (a) OM image of a triangular/hexagonal shape 2D template when the growth temperature is 650 °C. (b) OM image of a rectangular shape 2D template when the growth temperature is 800 °C. These images are

Changes to the manuscript.

- Figure R13 has been added as **Figure S13** in the revised SI
- Several sentences have been added on **Page 10**. “Note that MnN, r-NiN, and r-Co_xNi_yN show rectangular morphology, which is decided by the morphology of the template. For instance, as shown in Figure S13, 2D NiN exhibits a triangular morphology when synthesized at 650 °C, which corresponds to the triangular shape of the 2D template formed at that temperature. Conversely, when the step 1 temperature is increased to 800 °C, a

rectangular shape 2D template forms, resulting in rectangular 2D NiN flakes.”

Comment 3. It would be beneficial to include references for the sentence in lines 66-69.

We thank the reviewer for the suggestion, and the related reference (*Nat. Commun.* 16, 1623 (2025); *CRC Handbook of Chemistry and Physics*. 98th edn, (CRC Press, 2017).) has been added in the manuscript.

Changes to the manuscript.

- The sentence on Page 4 has been rewritten and two new references have been added in the revised manuscript “Transition metal chlorides (TMCl)s are therefore a potential choice due to their low conversion energy, while their high-temperature instability caused by their low sublimation temperature makes this process practically difficult.^{36,37}”

Comment 4. In Fig. 5a-f, the thicknesses are not directly displayed; line cuts are necessary.

Response 4. We thank the reviewer for this suggestion. We have updated Figure 5 (now Figure R14 in the response) to include AFM height profiles (line cuts) for each of the TMN flakes shown in panels (a)-(f), directly displaying their thicknesses.

Figure R14. Magnetism of 2D TMNs and their alloys. (a)-(f) Atomic force microscopy and MFM images of individual (a) 2D VN, (b) 2D CrN, (c) 2D MnN, (d) 2D Fe₂N, (e) 2D CoN, and (f) 2D NiN flakes measured at 300 K. The scale bar is 2 μm.

Changes to the manuscript.

- Figure R14 has been reproduced as Figures 5a-5f in the revised manuscript to replace the original Figures 5a-5f.

Response to Reviewer #3

Comment. The authors report a universal reverse-thermal-field CVD strategy for synthesizing two-dimensional transition metal nitrides (TMNs). In this approach, metastable metal chloride crystal templates are first deposited on mica substrates and then converted to TMNs by high-temperature nitridation, before the chloride templates are lost. Using this method, the authors demonstrate the growth of VN, CrN, MnN, Fe₂N, CoN, NiN, and their alloys. Structural and compositional characterizations (TEM, EDS, AFM) are presented, while magnetic properties are probed by MFM and SQUID. Overall, the technique represents a potentially viable route toward 2D TMNs and is therefore an important contribution to the 2D materials community. However, additional details and clarifications are needed:

Response. Thank you very much for your positive recommendations. We appreciate the reviewer by writing “universal”, “a potentially viable route” and “an important contribution to the 2D materials community”. Additional details and clarifications according to the reviewer have been added to the manuscript to strengthen our work.

Comment 1. A more detailed parametric study of the different materials should be conducted. For example, how was the temperature for Field 1 and 2 selected? Was any optimization performed in selecting these values? Can the crystal quality/defect density of the as-grown TMN be quantified? Such as using Raman peak width/position, and corroborated with XRD/XPS if feasible.

Response 1. We thank the reviewer for these important questions. **(1) Regarding temperature selection for Field I and II.** Taking CoN as an example, the operating high temperatures for Fields I and II (600-800 °C) were selected based on two key criteria derived from Figure R15 (Figure S4 in SI): (i) The thermogravimetric analysis (TGA) curves indicate that these temperatures are sufficient for the sublimation and deposition of the TMCl precursors to form 2D templates. (ii) The calculated Gibbs free energy (Figure R15b) confirms that the solid-to-solid conversion from TMCl to TMN becomes thermodynamically favorable within this temperature range. Further evolution of the growth results of 2D CoN under different growth temperatures was shown in Figure R16, which shows that when the temperature of Field II was

600 °C, some solid-state 2D CoN can be fabricated, while some products showed irregular shape, demonstrating that the conversion rate of deposited CoCl_2 might be similar to their sublimation rate, leading to incomplete conversion. Conversely, by increasing the temperature of the Field II to 750 °C, almost all the products showed regular triangular/hexagonal shape, which indicates that the conversion rate of deposited CoCl_2 was much greater than their sublimation rate at this temperature for completely solid-to-solid conversion.

(2) Regarding quantifying defect density in these nonlayered 2D materials. As metallic, non-layered materials with thicknesses typically between 10-20 nm and limited substrate coverage, conventional techniques like Raman spectroscopy (no characteristic peaks for metals), XRD, and XPS are ineffective due to the unavoidable substrate effect. Alternatively, we conducted HRTEM and EDS to study their structure and composition, to qualitatively reveal their crystal quality. The clear HRTEM images and SAED patterns of all the 2D TMNs and their alloys (inserts in Figures 17b, 17f, 17j, 18a, 18d and 18g) shown in the manuscript demonstrate their high crystallinity. As shown in Figures 18b, 18f and 18h, the metal elements are uniformly distributed among the flakes, demonstrating the element uniformity in the 2D TMN alloys.

Figure R15. Physical and chemical properties of transition metal chlorides. (a) TGA curves of the VCl_3 , CrCl_3 , CoCl_2 , FeCl_3 , MnCl_2 and NiCl_2 . (b) The Gibbs energy barrier of the solid-to-solid conversion from transition metal chlorides to their corresponding nitrides at different temperatures.

Figure R16. OM images of CoN with increased temperature from 600 °C to 800 °C.

Figure R17. Structural and composition characterization of 2D TMNs. (a) Top-view atomic structure of Fe_2N . (b) HRTEM image and SAED pattern (inset) of the synthesized 2D Fe_2N flake. (c) Low magnification TEM image, EDS element mapping of Fe and N atoms, and (d) EDS of the 2D Fe_2N flake. (e) Atomic structure of VN from the top view. (f) HRTEM image of the synthesized 2D VN flake. The inset is the corresponding SAED pattern. (g) Low magnification TEM image, elemental EDS mapping of V and N atoms, and (h) EDS of the 2D VN flake. (i) Atomic structure of MnN from the top view. (j) HRTEM image of the synthesized 2D MnN flake. The inset is the corresponding SAED pattern. (k) Low magnification TEM image, elemental EDS mapping of Mn and N atoms, and (l) EDS of the 2D MnN flake.

2D MnN flake. The inset is the corresponding SAED pattern. (k) Low magnification TEM image, elemental EDS mapping of Mn and N atoms, and (l) EDS of the 2D MnN flake.

Figure R18. Structural and composition characterizations of 2D TMN alloys. (a) HRTEM image and SAED pattern of the synthesized 2D $\text{Co}_{0.6}\text{Ni}_{0.4}\text{N}$ flake. (b) Low magnification TEM image, elemental EDS mapping of Co, Ni, and N atoms, and (c) EDS of the 2D $\text{Co}_{0.6}\text{Ni}_{0.4}\text{N}$ flake. (d) HRTEM image and SAED pattern of the synthesized 2D $\text{Co}_{0.3}\text{Fe}_{0.7}\text{N}$ flake. (e) Low magnification TEM image, elemental EDS mapping of Co, Fe and N atoms, and (f) EDS of the HRTEM image and SAED pattern of the synthesized 2D $\text{Co}_{0.3}\text{Fe}_{0.7}\text{N}$ flake. (g) HRTEM image and SAED pattern of the 2D $\text{Co}_{0.2}\text{Ni}_{0.1}\text{Fe}_{0.7}\text{N}$ flake. (h) Elemental EDS mapping of Co, Ni, Fe and N atoms and (i) EDS of the 2D $\text{Co}_{0.2}\text{Ni}_{0.1}\text{Fe}_{0.7}\text{N}$ flake.

Comment 2. Can the CVD parameters, such as precursor loading or growth duration, be used to systematically control layer thickness, nucleation density, and flake size? Since the authors describe these TMNs as 2D materials, please report the thickness statistics, which should typically be in the range of a few nm. Further optimization would be necessary to probe genuine 2D characteristics rather than bulk-like properties.

Response 2. We thank the reviewer for this comment. Regarding the question that whether the CVD parameters can control the growth results, we conducted a systematic study and found that the growth temperature and growth time control the lateral size and thickness of the flakes.

Taking CoN as an example, we performed a detailed study on the effects of growth temperature and growth time during step 1 (Figures R17 and R18). The thicknesses of the flakes was obtained from the AFM characterization, and the lateral size of the flakes were measured by the OM characterization. To better study the relationship between the growth temperature/time with the thickness/lateral size, we statistically analyzed more than 500 flakes grown under different growth conditions. As a result, (1) Both growth temperature and growth time exhibit a volcano-shaped relationship with the lateral size of the resulting 2D CoN flakes. Under the optimized conditions (a growth temperature of 750 °C and a growth time of 5 minutes), we have achieved 2D CoN flakes with an average lateral size of several tens of micrometers, with the largest flake reaching 51 μm (Figure R19). (2) By increasing the growth temperature and growth time, the thicknesses of the 2D CoN will increase. Under the optimized conditions (a growth temperature of 600 °C and a growth time of 2 minutes), we have achieved 2D CoN flakes with an average thickness of about 10 nm, with the thinnest flake of 1.03 nm (Figure R20).

Figure R19. The relationship between morphology of CoN and growth temperature. (a) OM images of CoN with increased temperature from 600 °C to 800 °C, and all the growth time were 5 min. (b) The thicknesses of CoN under different growth temperatures. The thickness was obtained by AFM measurements of 249 flakes. (c) The lateral sizes of CoN under different growth temperature. The lateral size was obtained by OM characterization.

Figure R20. The relationship between morphology of CoN and growth time. (a) OM images of CoN with increased growth time from 2 min to 20 min, and all the growth temperatures were 650 °C. (b) The thicknesses of CoN with different growth time. The thickness was obtained by AFM measurements of 293 flakes. (c) The lateral sizes of CoN with different growth time. The lateral size was obtained by OM characterization.

Figure R21. The OM image of 2D CoN with the lateral size of 51 μm .

Figure R22. The AFM image of the 2D CoN with the thickness of 1.03 nm.

Changes to the manuscript.

- Figure R19, Figure R20, Figure R21 and Figure R22 have been added as **Figure S10**, **Figure S11**, **Figure S12a** and **Figure S12b** in the revised SI.
- Several sentences have been added on **Page 10**. “Taking CoN as an example, the lateral size and thickness of the 2D flakes are determined by the growth temperature and duration of step 1 (Figure S10 and S11). The lateral size exhibits a volcano-like relationship with both increasing temperature and time, whereas the thickness generally increases. The largest CoN flake obtained thus far has a lateral size of 51 μm (Figure S12a), achieved at a growth temperature of 750 $^{\circ}\text{C}$ with a 5-minute growth time, while the thinnest flakes (~ 1.03 nm, Figure S12b) are obtained at 600 $^{\circ}\text{C}$ with a 2-minute growth time.”

Comment 3. Figure S2 shows the rapid loss of metastable CrCl_3 crystal within 40 s. For the RTF-CVD process, please provide the time-temperature profile. How rapid is the heating/cooling transition from 100-200 °C up to 600-800 °C? This is important for researchers to replicate the method.

Response 3. We thank the reviewer for emphasizing the importance of this detail. We have provided a photograph of the setup (Figure R23) and the detailed time-temperature profile (Figure R24). The key to the RTF process is the physical translation of the furnace on a sliding rail, which switches the thermal environments of Field I and Field II within **several seconds**. This rapid transition is crucial as it instantly brings the substrate (and the TMCl template on it) to the high nitridation temperature, preventing the sublimation and enabling the solid-state conversion. The detailed process has been illustrated in the **Experimental Section** in the revised manuscript, as shown below for researchers to replicate the method.

Figure R23. Photo of the setup used to achieve RTF process.

Figure R24. Temperature profile of the growth process of RTF method. (a) Temperature of the Field I, where the precursor located. (b) Temperature of the Field II, where the substrate located.

Changes to the manuscript.

- Figure R23 and Figure R24 have been added as **Figure S6** and **Figure S8** in the revised SI.
- Several sentences have been rewritten on **Page 7** in the revised manuscript. “Figures 1d and S6-S8 illustrate the setup of the furnace and the growth process of RTF method and traditional CVD method (see details in Experimental Section and Supporting Information).”
- Several sentences have been added on **Pages 19 and 20** in the revised manuscript. “*Synthesis of 2D TMNs and their alloys*: the 2D TMNs and their alloys were fabricated using the RTF method. The typical fabrication process was carried out in a movable furnace, as shown in Figure S6. The precursors were placed in a quartz boat located in the upstream region of the quartz tube (Field I), and the mica substrates were positioned in the downstream region (Field II). Argon (100 sccm) was used as the carrier gas throughout the process. The growth involved two distinct steps: Step 1 involved the deposition of the 2D TMCl templates. In this step, Field I (precursor zone) was moved to the heating center and

heated to the target temperature (600-800 °C) from room temperature in 30 minutes and maintained for 2 minutes. During this step, Field II (substrate zone), situated approximately 10 cm away from the heating center, remained at a lower temperature (100-200 °C). At the end of Step 1, the furnace was swiftly translated along the rail to position Field II (substrate zone) at the heating center, initiating Step 2 (the conversion step). Simultaneously with this movement, a mixture of NH₃ and H₂ was introduced and maintained for 10 minutes. Consequently, the temperature of Field II rapidly increased to 600-800 °C and was held for 5 minutes, while Field I cooled down to approximately 100-200 °C. Finally, the furnace was cooled to room temperature under an Ar atmosphere.”

Comment 4. For the alloyed TMNs, are the metal elements randomly incorporated in a single phase, or is there any short-range ordering/phase segregation? Can the elemental composition be tuned by adjusting precursor ratios?

Response 4. We thank the reviewer for these questions. (1) For the elemental distribution: EDS mapping (e.g., Figure R25 for Co_{0.6}Ni_{0.4}N) consistently shows a homogeneous distribution of the different metal elements at the nanometer scale, indicating the formation of a uniform alloy without macroscopic phase segregation. (2) As for the composition tuning: The composition of the alloy is influenced by the relative partial pressures of the different MCl_x species during step 1, which is in turn related to the precursor ratios. Taking the Co_xNi_yN as an example (Figures R25 and R26), when the weight ratio of CoCl₂ : NiCl₂ precursor is about 0 : 1, the product is pure 2D CoN; when the weight ratio of CoCl₂ : NiCl₂ precursor is about 1 : 1 the weight ratio of Co and Ni in the alloys is about 0.6 : 0.4; and when the CoCl₂ : NiCl₂ precursor is about 1 : 0, the product is pure 2D NiN. These results demonstrate that adjusting the weight ratio of the precursor can be used to further tune the element composition, which will be our future work.

Figure R25. Structural and composition characterization of $\text{Co}_{0.6}\text{Ni}_{0.4}\text{N}$. (a) HRTEM image and SAED pattern of the synthesized 2D $\text{Co}_{0.6}\text{Ni}_{0.4}\text{N}$ flake. (b) Low magnification TEM image, elemental EDS mapping of Co, Ni, and N atoms, and (c) EDS of the 2D $\text{Co}_{0.6}\text{Ni}_{0.4}\text{N}$ flake.

Figure R26. Evolution of the atomic ratio of Co : Ni in the alloys with the weight ratio of $\text{CoCl}_2 : \text{NiCl}_2$ and OM images of products with different precursors.

Comment 5. The AFM and MFM images of the alloyed TMN, along with the corresponding thickness measurements, should be included in Figure 5. Does the magnetic property change with thickness?

Response 5. We thank the reviewer for this suggestion. (1) We have provided the corresponding thickness measurements in Figure 5 (Figure R27). (2) Regarding the relationship between the magnetic property and the thickness, we take the CoN as an example and conduct MFM characterization to reveal its magnetic behavior with different thicknesses. As shown in Figure R28, the MFM contrast of 2D CoN changes with thickness, indicating an evolution in magnetic properties from weak signals to strong signals with the increased thickness. Therefore, to ensure a comparison of intrinsic material properties in the main text, the SQUID measurements in Figure R25 (g-i) were performed on flakes with comparable thicknesses (~20 nm).

Figure R27. Magnetism of 2D TMNs and their alloys. (a)-(f) Atomic force microscopy and MFM images of individual (a) 2D VN, (b) 2D CrN, (c) 2D MnN, (d) 2D Fe₂N, (e) 2D CoN, and (f) 2D NiN flakes measured at 300 K. The scale bar is 2 μm. (g) M-H loops of the 2D TMNs and (h) their alloys with the magnetic field applied out of plane. The results are measured by SQUID at 10 K. The effect of mica substrate has been eliminated (Figure S19). (i) Statistics of H_c, M_s and Curie temperature (T_c)/Neel temperature (T_N) of the 2D TMNs and their alloys in (g) and (h). The T_c/T_N of each material is obtained from the M-T curves shown in Figure S20 and Figure S21. The materials with one, two, three and four types of metal elements are highlighted in yellow, pink, green and blue, respectively. And increasing H_c drives the transition from soft to hard magnetic materials.

Figure R28. The thickness dependent magnetism of 2D CoN. (a) AFM and (b) corresponding MFM images of 2D CoN flakes with thicknesses of 14.7 nm, 26.3 nm, 30.3 nm and 50.7 nm.

Changes to the manuscript.

- Figure R27 has been reproduced as **Figure 5** in the revised manuscript to replace the original Figure 5.